# Enapotamab Vedotin, an AXL-Specific Antibody-Drug Conjugate, Demonstrates Antitumor Efficacy in Patient-Derived Xenograft Models of Soft Tissue Sarcoma

**DOI:** 10.3390/ijms23147493

**Published:** 2022-07-06

**Authors:** Britt Van Renterghem, Agnieszka Wozniak, Patricia Garrido Castro, Patrick Franken, Nora Pencheva, Raf Sciot, Patrick Schöffski

**Affiliations:** 1Laboratory of Experimental Oncology, Catholic University of Leuven, 3000 Leuven, Belgium; britt.vanrenterghem@kuleuven.be (B.V.R.); agnieszka.wozniak@kuleuven.be (A.W.); 2Genmab, 3584 Utrecht, The Netherlands; pgc@genmab.com (P.G.C.); pfr@genmab.com (P.F.); npe@genmab.com (N.P.); 3Department of Pathology, University Hospitals Leuven, Catholic University of Leuven, 3000 Leuven, Belgium; raf.sciot@kuleuven.be; 4Department of General Medical Oncology, University Hospitals Leuven, 3000 Leuven, Belgium

**Keywords:** enapotamab vedotin, HuMAX-AXL-ADC, antibody-drug conjugate, AXL, patient-derived xenografts, soft tissue sarcoma

## Abstract

Doxorubicin (doxo) remains the standard of care for patients with advanced soft tissue sarcoma (STS), even though response rates to doxo are only around 14% to 18%. We evaluated enapotamab vedotin (EnaV), an AXL-specific antibody-drug conjugate (ADC), in a panel of STS patient-derived xenografts (PDX). Eight models representing multiple STS subtypes were selected from our STS PDX platform (n = 45) by AXL immunostaining on archived passages. Models were expanded by unilateral transplantation of tumor tissue into the left flank of 20 NMRI *nu*/*nu* mice. Once tumors were established, mice were randomized into an EnaV treatment group, or a group treated with isotype control ADC. Treatment efficacy was assessed by tumor volume evaluation, survival analysis, and histological evaluation of tumors, and associated with AXL expression. EnaV demonstrated significant tumor growth delay, regression, and/or prolonged survival compared to isotype control ADC in 5/8 STS PDX models investigated. Experimental passages of responding models were all found positive for AXL at varying levels, but no linear relationship could be identified between the level of expression and level of response to EnaV. One model was found negative for AXL on experimental passage and did not respond to EnaV. This study provides a preclinical rationale for the evaluation of AXL-targeting ADCs in the treatment of AXL-expressing sarcomas.

## 1. Introduction

Soft tissue sarcomas (STS) are a heterogeneous group of malignant tumors from mesenchymal origin, accounting for approximately 1% of adult and 10% of childhood malignancies [1,2]. Although relatively rare and potentially curable by surgical resection when localized, the need for more effective therapies for patients with inoperable or metastatic disease is highlighted by poor 5-year survival of only 15% [3]. Ever since the early 1970s, doxorubicin (doxo) monotherapy has remained the standard-of-care as the first agent to show meaningful activity in patients with advanced disease, regardless of the STS subtype [4]. However, according to recent large, randomized trials, objective responses to doxo are seen in only 14 to 18% of patients [5,6,7,8]. Meanwhile, patients treated with doxo are exposed to a number of risks, including dose-dependent and potentially irreversible cardiotoxicity which limits the cumulative dose that can be administered over the lifetime of a patient [2,9].

The recent introduction of advanced molecular techniques (i.e., fluorescence in situ hybridization (FISH), reverse transcription-polymerase chain reaction (RT-PCR), and methods based on next-generation sequencing) has dramatically improved our understanding of important oncogenic pathways in sarcoma and even led to the identification of several actionable targets and alternative treatment options for these patients [10]. Aberrant expression of the receptor tyrosine kinase AXL has recently been reported in a variety of cancers, including the more common STS subtypes such as leiomyosarcoma and liposarcoma [11,12,13]. According to publicly available gene expression data from the Cancer Genome Atlas (TCGA) regarding treatment-naïve tumors, sarcomas in general have the highest intrinsic AXL expression among all cancer types. In gastrointestinal stromal tumors (GIST) and other cancer types, AXL overexpression has been associated with intrinsic or acquired resistance to targeted therapies, as well as to chemotherapy and radiotherapy [14,15,16,17].

Enapotamab vedotin (EnaV, also known as HuMAX-AXL-ADC) is an AXL-specific human IgG1 antibody conjugated to the microtubule disrupting agent monomethyl auristatin E (MMAE) through a protease cleavable valine-citrulline (vc) linker [18]. Once EnaV binds to AXL expressed on the cell membrane of AXL-positive tumor cells, the complex is internalized and cleaved by lysosomal proteases, releasing free MMAE. As a result, MMAE can diffuse freely within the cell where it binds to the microtubules and inhibits tubulin polymerization. Thereby MMAE interferes with the proper assembly of the mitotic spindle during cell division, resulting in cell cycle arrest and eventually cell death. Since free MMAE can also diffuse back out of the cells, EnaV has the potential to cause additional bystander effects by killing surrounding AXL-negative tumor cells [18].

Early studies with EnaV have demonstrated potent antitumor activity that was associated with AXL expression in patient-derived xenograft (PDX) models derived from various tumors, such as melanoma, lung, pancreatic and cervical cancers [18,19,20]. In the present in vivo study, we evaluated the efficacy of EnaV in STS PDX and explored the potential of AXL expression as a predictive biomarker to support further development of AXL-targeted therapies in this setting.

## 2. Results

### 2.1. Evaluation of AXL Expression on the STS PDX Tissue Microarrays

STS PDX tissue microarrays containing tumor tissue from all available models (n = 45) were evaluated for AXL immunopositivity. Eight models with a median AXL H-score >120 were selected to evaluate the antitumor efficacy of EnaV (Figure 1), including dedifferentiated liposarcoma (UZLX-STS3^DDLPS^, -STS124^DDLPS^, and -STS204^DDLPS^), leiomyosarcoma (UZLX-STS81^LMS^ and -STS128^LMS^), myxofibrosarcoma (UZLX-STS126^MFS^ and -STS132^MFS^), and undifferentiated pleomorphic sarcoma (UZLX-STS84^UPS^). Apart from their high AXL expression, these models represent the more common histological subtypes of STS [1]. While the majority of models showed relatively stable AXL expression over included passages, several models (i.e., UZLX-STS128^LMS^) showed pronounced inter-passage variability. For this reason, the AXL expression for comparison with the response to EnaV in a given model was evaluated on experimental passage by IHC on isotype control ADC-treated tumors. Of note, models UZLX-STS3^DDLPS^ and -STS84^UPS^ have been used previously for in vivo drug testing [21,22].

### 2.2. Characterization of the Selected STS PDX Models

Despite small changes in the extent of tumor necrosis, myxoid areas and/or cellularity, original patient tumors, and PDX tumors shared the same characteristic morphological and molecular features (Figure 2). UZLX-STS3^DDLPS^, -STS124^DDLPS^, and -STS204^DDLPS^ consisted of spindle and epithelioid shaped cells with typical nuclear MDM2 expression and *MDM2* amplification. UZLX-STS81^LMS^ and -STS128^LMS^ demonstrated pleomorphic cell morphology with cytoplasmic alpha-SMA expression. UZLX-STS126^MFS^ and -STS132^MFS^ showed alternation of hypercellular and hypocellular myxoid areas, of which the latter have become more dominant over passages in both xenografts. UZLX-STS84^UPS^ showed pleomorphic cell morphology with diffuse areas of necrosis.

### 2.3. Antitumor Activity of EnaV in the Selected STS PDX Models

Two weeks after the last treatment (day 22), EnaV-treated tumors showed significantly delayed tumor growth compared to isotype control ADC as determined by unpaired t-test in the UZLX-STS3^DDLPS^, -STS124^DDLPS^, -STS128^LMS^, and -STS84^UPS^ xenografts, with significant tumor regression compared to baseline as determined by a paired t-test in UZLX-STS84^UPS^ and -STS128^LMS^ (Figure 3A and Appendix A). By the end of tumor volume evaluation (max. 100 days), we observed complete tumor regressions in all remaining EnaV-treated animals of UZLX-STS128^LMS^ and all but one of UZLX-STS84^UPS^, and a persistent tumor growth delay in UZLX-STS3^DDLPS^ and -STS124^DDLPS^ (Figure 3B). Individual relative tumor growth curves are provided in Appendix A. Survival analysis on Kaplan-Meier curves showed significantly prolonged survival of EnaV-treated mice compared to isotype control ADC in the UZLX-STS84^UPS^ and -STS126^MFS^ xenografts (Figure 3C). Based on the above-mentioned criteria, models UZLX-STS84^UPS^, -STS128^LMS^, -STS3^DDLPS^, -STS124^DDLPS,^ and -STS126^MFS^ were categorized as responding and UZLX-STS132^MFS^, -STS204^DDLPS^, and STS3^DDLPS^ as non-responding.

Both treatments were well tolerated in mice based on general well-being. Mice that were sacrificed because of body weight loss or found dead were equally distributed over both treatment arms (Appendix A). Only in UZLX-STS84^UPS^, an increased mortality of animals was observed in the EnaV-treated arm, most likely as a consequence of infection with mouse hepatitis virus that was detected during the follow-up period of this experiment. Mice of the isotype control ADC-arm were already sacrificed by then as all tumors had reached the maximum tumor volume. Additionally, one mouse was sacrificed during active treatment with EnaV (day 8) because of body weight loss >18%. The relative body weight evolution of mice in each experiment is shown in Appendix A.

### 2.4. Histological Assessment of PDX Tumors

On the day of tumor collection (day 22, 3 mice/group), isotype control ADC-treated tumors from all histologically evaluable models demonstrated high mitotic count characteristic of the aggressive behavior of these tumors, with >10 mitotic figures per 10 HPF (Figure 4A,B). EnaV-treated tumors of UZLX-STS3^DDLPS^, -STS124^DDLPS^, -STS204^DDLPS^, -STS126^MFS^, and -STS132^MFS^ showed a slightly reduced number of mitotic and increased number of apoptotic cells compared to isotype control ADC (Figure 4C,D). Tumors of UZLX-STS84^UPS^ and -STS128^LMS^ could not be assessed as EnaV-treated tumors were too small to be evaluated by 10 HPF. Representative scans showing extensive necrosis with few viable cells left in response to EnaV treatment are provided in Appendix A. These results are in line with the antitumor responses observed in these models.

### 2.5. AXL Expression as Potential Predictive Biomarker

As several models showed variating levels of AXL expression from passage to passage on tissue microarray (Figure 1), the AXL expression for comparison with the response to EnaV in a given model was evaluated on experimental passage (i.e., isotype control ADC-treated tumors) by AXL IHC-score (Table 1 and Figure 2). Interestingly, the only model that was found to be negative on the experimental passage (UZLX-STS81^LMS^) was also categorized as non-responding. Meanwhile, all EnaV-responding models showed AXL expression with scores ranging from one to three, but no linear relationship could be identified between the level of target expression and the level of response to EnaV. For example, UZLX-STS84^UPS^ that showed response to EnaV with significant tumor growth delay, tumor regression, and prolonged survival had an AXL IHC-score of only one, while UZLX-STS3^DDLPS^ that responded with only significant tumor growth delay had a score of three. These results support EnaV’s mechanism of action (MoA), where some level of AXL expression is required as an ‘address’ to deliver its cytotoxic payload, but they also suggest that additional factors contribute to the actual response to EnaV downstream of AXL.

## 3. Discussion

First introduced in the 1970s, doxo was the first anticancer agent to produce meaningful response rates in patients with advanced STS [4]. Over the following decades, many other cytotoxic agents have been investigated in this setting [23]. While only a few received marketing authorizations as second-line treatments (e.g., eribulin, trabectedin, and pazopanib), none of these agents succeeded to surpass the limited activity of single-agent doxo nor replace it as a first-line standard-of-care [24,25,26]. Although often promising in early phase studies, efficacy signals seem to get lost in the highly heterogeneous cohorts of large phase III trials [23]. A newer generation of clinical trials is focusing on histologically and/or molecularly defined subgroups of STS with promising results [27]. Given the common overexpression of AXL in sarcomas as compared to other solid tumors and considering its increased expression in cancers with resistance to chemotherapy or radiotherapy, as well as its association with epithelial-to-mesenchymal transition in solid tumors [14,15,16,17], AXL is a rational candidate for targeted treatment approaches in mesenchymal malignancies. We evaluated EnaV, an AXL-specific ADC, in sarcoma PDX models and explored the potential of AXL expression as a predictive biomarker to support further development of AXL-targeted therapies in this setting.

As an antibody-drug-conjugate (ADC) that is designed to combine the targeting capacity of a monoclonal antibody with the toxic properties of an established cytotoxic payload, EnaV is only using AXL as an ‘address’ to deliver its payload but does not compete with its ligand for binding [18]. As such, EnaV can exert its antineoplastic activity independently of AXL- signaling mediated tumorigenesis. We believe this approach has advantages over the use of selective AXL signaling inhibitors in sarcomas that are displaying complex karyotypes (>80% of STS) and are potentially driven by multiple oncogenic pathways [10]. At this moment, at least four ADCs are under clinical investigation in sarcoma addressing targets such as AXL, neural cell adhesion molecule (NCAM), receptor tyrosine kinase-like orphan receptor (ROR), and many others are being evaluated in preclinical studies [28].

For the current study, models were selected from our extensive “XenoSarc” platform of well-characterized human sarcoma xenografts [29]. Tissue microarrays containing archived tumor tissue from all available models (n = 45) were stained and evaluated for AXL expression. Eight models with strong AXL expression (median AXL H-score > 120) were selected, representing the more common STS subtypes of dedifferentiated liposarcoma, leiomyosarcoma, myxofibrosarcoma, and undifferentiated pleomorphic sarcoma. We have previously shown these models can be used for in vivo preclinical drug testing [21,30,31,32,33,34,35], and can successfully predict the outcome of a clinical trial [22,36].

In 5/8 STS PDX models investigated, EnaV induced a significant response in terms of tumor growth delay, tumor regression, and/or improved survival compared to isotype control ADC. Although only two models responded with significant tumor shrinkage (i.e., the gold standard of response in oncology), complete tumor regressions could be observed in all but one PDX tumors by the end of both experiments. An additional two models responded with limited, although significant, tumor growth delay (Table 1); however, it should be noted that tumor growth stabilization or delay is already considered a clinically meaningful response in this setting given the low objective response rates observed with doxorubicin [37]. For this reason, it would have been interesting to include a doxorubicin-treated group in these experiments for direct comparison to the first-line standard-of-care, as done previously [21,22,33,38].

All EnaV-responding models were positive for AXL on experimental passage, but the level of AXL expression as determined by the AXL IHC-score was not necessarily in linear relation with the level of response to EnaV. The only model found to be negative on experimental passage (UZLX-STS81^LMS^ p.17) was also non-responding to EnaV. These results support that EnaV’s activity is primarily determined by the presence of AXL expression but suggest that additional factors contribute to the response to EnaV downstream of AXL, e.g., the inherent sensitivity of the tumor cells to the payload as demonstrated in previous work [18]. Of note, UZLX-STS81^LMS^ was initially selected as an AXL-expressing model when analyzed on tissue microarrays containing earlier passages of the models. The differential expression of AXL between those passages in the same xenograft may be attributed to the clonal dynamics that take place over serial passaging. Interestingly, such clonal selection occurring in xenograft models over passaging seems to reflect the tumor progression of the donor patients, making these models even more relevant [39,40]. EnaV was well tolerated based on the general well-being in mice, but no extensive toxicological analysis was performed (e.g., histopathological evaluation of organs) as this was beyond the scope of this work, although this has been addressed in other work [18].

In conclusion, we observed in vivo antitumor activity of EnaV in 5/8 STS PDX models investigated at a well-tolerated dose of only 4 mg/kg. While 3/5 EnaV-responding models demonstrated either limited tumor growth delay or prolonged survival, two responded with remarkable tumor shrinkage resulting in complete tumor regressions. Moreover, AXL expression was associated with the response upon EnaV in the xenografts, although this response was not necessarily linear with the level of expression. The present study provides the preclinical rationale for further clinical investigation of AXL-targeted therapies in patients with AXL-expressing sarcomas.

## 4. Methods and Materials

### 4.1. Patient-Derived Sarcoma Xenograft Models

Xenograft models used in this study were selected from our extensive “XenoSarc” platform of well-characterized STS PDX models, available at the Laboratory of Experimental Oncology, Catholic University of Leuven (KU Leuven), Leuven, Belgium [29]. Xenografts were established by subcutaneous transplantation of fresh tumor tissue from consenting STS patients in NMRI *nu*/*nu* mice (Janvier Labs, Le Genest-Saint-Isle, France) as previously described [29].

For model selection, previously constructed PDX tissue microarrays containing archived tumor tissue from all available models (n = 45) were immunostained for AXL (clone C89E7; Cell Signaling Technology, Danvers, MA, USA, cat. no #8661) on a Ventana Discovery Autostainer (Ventana Medical Systems, Oro Valley, AZ, USA). Detection was performed using anti-Rb HQ (Roche, Basel, Switzerland, cat. no #7017936001) and anti-HQ HRP (Roche, cat. no #7017812001), followed by diaminobenzidine (DAB) precipitation (Roche, cat. no. #5266645001). For each model, multiple passages were included into the tissue microarrays with 1–3 cores per passage covering the center and edges of the tumor tissue, which is important considering the heterogeneous expression of most targets. Tumor tissue cores were scored for AXL by a certified pathologist, categorizing individual cells as negative, weak (AXL 1+), moderate (AXL 2+), or strong (AXL 3+) by visual analysis. Out of these scores, the AXL H-score was calculated for every passage using the following formula: (1 × % AXL 1 + tumor cells) + (2 × % AXL 2 + tumor cells) + (3 × % AXL 3 + tumor cells). Models with high AXL H-score over multiple passages (median AXL H-score > 120) were selected for the in vivo study regardless of the STS subtype.

### 4.2. Drugs and Reagents

EnaV and isotype control ADC (IgG1-b12-vcMMAE) were provided by Genmab BV (Utrecht, The Netherlands) as a stock solution (10 and 9.80 mg/mL, respectively) and diluted to a working solution of 0.8 mg/mL in phosphate buffered saline (PBS) immediately before injection. The following antibodies were used for manual immunohistochemistry (IHC) of experimental PDX tumors: alpha-smooth muscle actin (alpha-SMA; Agilent Technologies, Santa Clara, CA, USA, cat. no #M085129-2), AXL (Cell Signaling Technology, cat. no #8661), cleaved poly (ADP-ribose) polymerase (cleaved PARP; Abcam, Cambridge, UK, cat. no #32064), murine double minute 2 homolog (MDM2; ThermoFisher Scientific, Waltham, MA, USA, cat. no #337100) and phospho-histone H3 (pHH3; Cell Signaling Technology, cat. no #9701). All sections were incubated using Envision-HRP-anti-rabbit/mouse (Agilent Technologies, cat. no #K4003/#K4001), except for cleaved PARP for which SignalStain Boost IHC Detection Reagent (Cell Signaling Technology, cat. no #8125S) was used. Subsequently, stainings were developed using DAB (Agilent Technologies, cat. no #K346811), followed by hematoxylin counterstaining (VWR, Radnor, PA, USA). The following probes were used for MDM2 FISH: MDM2 (12q15)/SE12 (Leica Biosystems, Deer Park, IL, USA). Tissue pretreatment, hybridization, and detection were carried out according to the manufacturer’s instructions.

### 4.3. Experimental Setup

For every xenograft model selected, experimental cohorts were created by transplantation of fresh tumor fragments (2-4 mm diameter) into the left flank of 20 female NMRI *nu*/*nu* mice (aged 7–8 weeks). Once the majority of tumors reached a volume of 100–150 mm^3^, mice were randomized (day 0) to EnaV (4 mg/kg) or isotype control ADC (4 mg/kg) treatment. On days 1 and 8 of each experiment, mice received intravenous tail vein injection with 5 mL/kg of the respective working solution. During the experiment, tumors and body weight were measured three times per week. Tumors were measured three-dimensionally by caliper and volumes were calculated using the following formula: tumor volume (mm^3^) = length (mm) × width (mm) × height (mm). The experiment ended for individual mice when the tumor volume reached >2000 mm^3^, tumors started ulcerating, or the mouse showed body weight loss >18%, with exception of three mice per group that were sacrificed earlier for histological evaluation of tumors. Experiments were terminated at a maximum of 100 days. Responses to EnaV were defined as significant tumor growth delay, tumor regression, and/or prolonged survival compared to isotype control ADC. A detailed description of the number of mice/tumors included in each experiment can be found in Appendix A.

### 4.4. Histological Assessment of PDX Tumors

Three mice per group were randomly selected and sacrificed on day 22 (i.e., two weeks after the last treatment) for tumor collection in 4% formaldehyde in PBS. Formalin-fixed tumors were embedded in paraffin and cut into 4µm-thick sections to confirm model characteristics and to evaluate the treatment efficacy by means of mitotic and apoptotic activity. H&E staining was used for counting mitotic and apoptotic cells in 10 high power fields (HPF) at 400-fold magnification (0.45-mm field diameter). pHH3 and cleaved PARP staining, markers for proliferative and apoptotic activity, were used for counting the number of immune-positive tumor cells in 10 HPF. Additionally, the AXL expression for comparison with the response to EnaV in a given model was evaluated on experimental passage by IHC on isotype control ADC-treated tumors. Whole tumor sections were given an AXL IHC-score based on the overall intensity of staining: 0—negative, 1—weakly, 2—intermediate, and 3—strongly positive. Histological analysis was performed using a CH30 microscope (Olympus, Tokyo, Japan). Pictures were taken using a BX43 light microscope (Olympus).

### 4.5. Statistics

Changes in tumor volumes relative to baseline were compared between both treatment arms. The unpaired t-test was applied to identify significant differences in the average relative tumor volume between both treatment arms (i.e., tumor growth delay) on day 22. The paired t-test was applied to distinguish significant tumor regression from tumor growth delay by comparing tumor volumes within the same treatment arm at two different time points (day 1 vs. day 22). The log-rank test was applied for comparison of the survival distribution between the two treatment arms on Kaplan-Meier curves. Animals that were sacrificed because of body weight loss >18%, found dead, or reached the end of observation were censored on the curve. Mice that were sacrificed for histological assessment on day 22 were excluded from Kaplan–Meier analyses but included in tumor volume evaluation. GraphPad Prism software (version 8.4.3, San Diego, CA, USA) was used for all graphical presentations and calculations. A *p*-value < 0.05 was considered statistically significant.

## Figures and Tables

**Figure 1 ijms-23-07493-f001:**
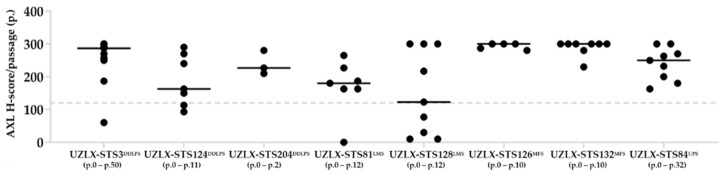
AXL H-scores of individual passages (p.) and included p. ranges for selected models.

**Figure 2 ijms-23-07493-f002:**
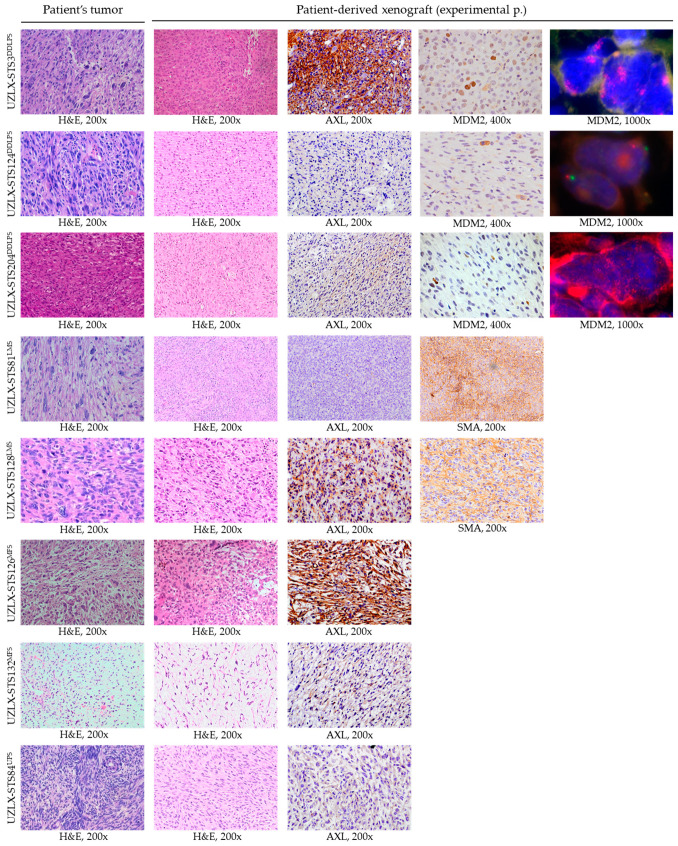
Characterization of the patient-derived sarcoma xenograft models used in this study. Representative images of H&E, immunostainings and FISH of the original patient tumors and the corresponding patient-derived xenografts. alpha-SMA: alpha smooth muscle actin; H&E: hematoxylin & eosin; MDM2: mouse double minute 2 homolog; p.: passage; 200×: 200-fold magnification; 400×: 400-fold magnification; 1000×: 1000-fold magnification.

**Figure 3 ijms-23-07493-f003:**
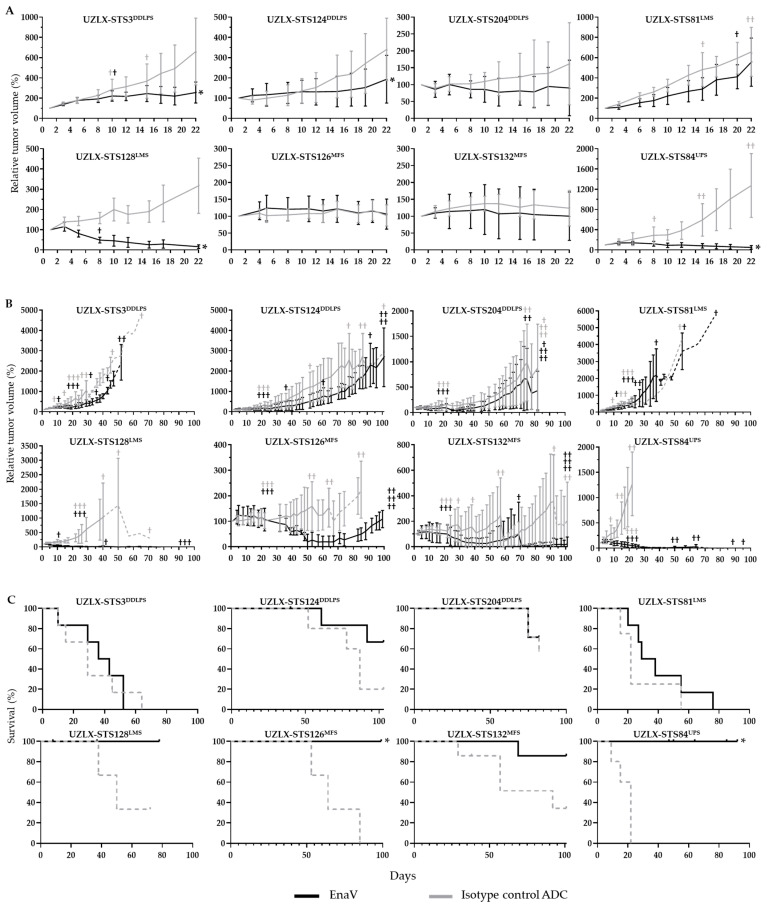
Tumor volume evaluation curves and survival curves of all sarcoma patient-derived sarcoma xenografts included in this study: (**A**) average relative tumor volume ± standard deviation (%) until day 22 and (**B**) average relative tumor volume (%) until day 100. Statistical significance as determined by unpaired t-test. Dotted lines represent data from less than three animals. Number of †: number of mice sacrificed during the experiment. (**C**) Kaplan-Meier curves with statistical significance as determined by log-rank test. Animals that were sacrificed because of body weight loss >18%, found dead or that reached the end of observation were censored. Mice sacrificed for histological evaluation day 22 (3 mice/group) were included in tumor volume evaluation but excluded from survival analysis. * *p* < 0.05 compared to isotype control ADC.

**Figure 4 ijms-23-07493-f004:**
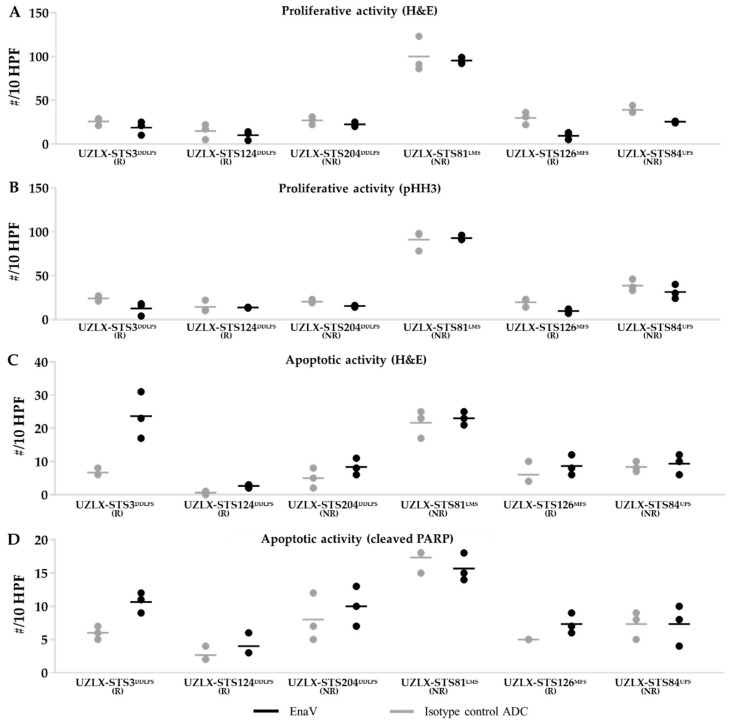
Assessment of mitotic and apoptotic activity of tumors (n = 3) collected on day 22. Mitotic cell count assessed on (**A**) H&E and (**B**) pHH3 staining. Apoptotic cell count assessed on (**C**) H&E and (**D**) cleaved PARP staining. Data are presented as average ± standard deviation. Samples from UZLX-STS84^UPS^ and -STS128^LMS^ could not be assessed, as EnaV-treated tissues were too small to count 10 HPF. HPF: high power fields; H&E: hematoxylin & eosin; pHH3: phospho-histone H3; PARP: poly (ADP-ribose) polymerase; R: responding model; NR: non-responding model.

**Table 1 ijms-23-07493-t001:** The AXL IHC-score and tumor growth response to EnaV-treatment of all sarcoma xenografts included in this study.

Xenograft	Passage (p.)	AXL IHC-Score	Response to EnaV
UZLX-STS84^UPS^	p.28	1	Response ^TGD, TR, S^
UZLX-STS128^LMS^	p.6	2	Response ^TGD, TR^
UZLX-STS3^DDLPS^	p.5	3	Response ^TGD^
UZLX-STS124^DDLPS^	p.18	1	Response ^TGD^
UZLX-STS126^MFS^	p.17	3	Response ^S^
UZLX-STS132^MFS^	p.6	1	No response
UZLX-STS204^DDLPS^	p.5	1	No response
UZLX-STS81^LMS^	p.17	0	No response

TGD: significant tumor growth delay compared to isotype control; TR: significant tumor regression compared to isotype control; S: significantly prolonged survival compared to isotype control ADC.

## Data Availability

The datasets used and/or analyzed during the current study are available from the corresponding author on reasonable request.

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
