# Peer review of "Enapotamab Vedotin, an AXL-Specific Antibody-Drug Conjugate, Demonstrates Antitumor Efficacy in Patient-Derived Xenograft Models of Soft Tissue Sarcoma"

_ijms, 2022, doi:10.3390/ijms23147493_

Round 1

Reviewer 1 Report

Manuscript entitled "Enapotamab Vedotin, an AXL-Specific Antibody-Drug Conjugate, Demonstrates Antitumor Efficacy in Patient-Derived Xenograft Models of Soft Tissue Sarcoma"

Major issues:

1. For MFS cases, the representative sections containing myxoid areas should be shown for patients' samples.

2. For DDLPS samples, The MDM2 is not convincingly stained. CDK4 should be added.

3. The authors have to show the treatment effects in post-treated PDX samples. Did the treatment impacts cell proliferation? (Ki-67 should be added); angiogenesis? (CD31 should beadded); or lead to necrosis or apoptosis? In short, the mechanism should be disclosed in cell and tumor sample levels. 

Author Response

Major issues:

  1. For MFS cases, the representative sections containing myxoid areas should be shown for patients' samples.

We retrieved myxofibrosarcoma patient samples and replaced pictures of H&E in Figure 2, showing now more representative myxoid parts of the tumors. Additionally, new pictures of H&E were taken for PDX samples of UZLX-STS126, as well as for UZLX-STS3 according to comments from reviewer #2.

  1. For DDLPS samples, The MDM2 is not convincingly stained. CDK4 should be added.

Even though it is possible that the level of MDM2 expression could vary in different passages, we agree that MDM2 immunohistochemistry alone might not be convincing enough for model confirmation. For this reason, MDM2 FISH is performed every few passages for all DDLPS PDX model of our platform (see our publication on the platform for methods and data: https://doi.org/10.1158/1535-7163.MCT-18-1045). We added images of MDM2 amplification by FISH performed on experimental passages of UZLX-STS3, -STS124 and –STS204 to Figure 2.

  1. The authors have to show the treatment effects in post-treated PDX samples. Did the treatment impacts cell proliferation? (Ki-67 should be added); angiogenesis? (CD31 should be added); or lead to necrosis or apoptosis? In short, the mechanism should be disclosed in cell and tumor sample levels.

Post-treated PDX samples were histologically evaluated by counting the number of mitotic cells on H&E and pHH3 (a specific marker for mitosis found to more reliable than Ki67, see for example https://doi.org/10.1097/PAI.0000000000000555 or https://doi.org/10.18632/oncotarget.17775) and apoptotic cells on H&E and cleaved PARP in 10 HPF. Of note, H&E based assessment is considered a gold standard for both mitotic and apoptotic count. As illustrated in Figure 4, EnaV-treated samples showed slightly decreased mitosis and increased apoptosis as the response to treatment. Samples of UZLX-STS84 and –STS128 could not be evaluated as they were too small to identify 10 HPF. Nevertheless, we now added representative scans (Figure S3) showing extensive necrosis with only few viable cells left as response to EnaV. The following paragraph has been changed in the manuscript (p.8):

“EnaV-treated tumors of UZLX-STS3DDLPS, -STS124DDLPS, -STS204DDLPS, -STS126MFS and -STS132MFS showed a slightly reduced number of mitotic and increased number of apoptotic cells compared to isotype control ADC (Figure 4C & 4D). Tumors of UZLX-STS84UPS and -STS128LMS could not be assessed as EnaV-treated tumors were too small to be evaluated by 10 HPF. Representative scans showing extensive necrosis with few viable cells left as response to EnaV treatment are provided in Supplementary Figure S3. These results are in line with the antitumor responses observed in these models.”

Reviewer 2 Report

 In this manuscript, the authors present in vivo study demonstrating the efficacy of AXL targeting ADC in soft tissue sarcoma PDX models. The authors concluded that “In 5/8 STS PDX models investigated, EnaV induced a significant response in terms of tumor growth delay, tumor regression, and/or improved survival compared to isotype. The results also show that AXL expression is necessary for the efficacy of this treatment but not significantly associated with response magnitude, indicating that other tumor characteristics are also involved in the overall results of this ADC treatment.

Unfortunately, the in vivo studies were not powered and therefore the significance of the results was not reached in the majority of the analyses.  Is the tumor really a responder with minimal differences in tumor volume between control and treated groups (not significant) and no survival benefits? Making conclusions based on no significant changes (tumor progression inhibition or survival) seems like an overinterpretation of the data and does not seem appropriate.

Comments:

1)     Once the majority of tumors reached a volume of 100-150mm3, mice were randomized. Information about the range of tumor sizes should be included. Also. the criteria for selection of the three tumors per group for sacrifice at 22 days selected should be included- smaller or larger at enrolment, or size at 22 days?

2)     The authors state that “The experiment ended for individual mice when the tumor volume reached >2000 mm3 or the mouse showed body weight loss >18%, with exception of three mice per group that were sacrificed earlier for histological evaluation of tumors.” However, based on the individual tumor growth curves it does not appear that these criteria were consistently used. For example, UZLX_STS126 all mice were sacrificed before reaching either one of these criteria, while UZLX-STS124 mice with control tumors were sacrificed at the tumor size of ~4000 mg.

3)     The AXL expression on TMA was presented as an IHC score based on immunoreactivity intensity and percentage of cells at each intensity; while the AXL expression in the study tumors was stated as “based on the overall intensity of the staining: 0 - negative, 1 - weakly, 2 - intermediate and 3. Why were different criteria used?

4)     Figure 1 shows the H&E of the parental tumors and the PDX models and PDX IHC. It is stated: “Original patient tumors and PDX tumors were compared and shared the same characteristic morphological features.”  However, based on images in figure 1 the morphologies of the patient and PODX tissues do not seem to have similar morphology. E.g., STS3 STS126. Moreover, the H&E and AXL IHC of STS126 do not look like the same tumor. Minor comment: STS3 MDM2 IHC does not appear to be 200x

5)     The authors state: “Additionally, a non-significant trend for tumor growth delay could be observed in the EnaV-treated tumors of the more slowly growing models STS126MFS, -STS132MFS.” Looking at the data provided, the tumors really did not grow within the first 22 days and therefore no inhibition of tumor progression was observed.  It does not seem appropriate to evaluate tumor inhibition when the control tumors themselves are not growing. However, if one would be looking at day 60, the control tumor might have doubled, then this tumor could be a responder with the development of acquired resistance (day >80) (of course the concussion would be possible only if the results reach significance in powered study).

6)     A similar comment is valid for the survival analyses. No conclusions can be made about potential survival benefits if the differences are not significant.

7)     The difficulty of a correct interpretation is rather visible when one compares the 22-day tumor response of STS3, which is marked as a significant response, growth curves for the whole study of this model that does not show much inhibition, and survival analysis where the lines of control and treated groups show no differences.

8)     In figure 4A tumors that did not show generally any increase in tumor volume during the 22 days period (126 and 132) have similar #/10HPF as tumors that grew relatively fast (STS3 and STS81). Moreover, the 126 showed decreased proliferation in the treatment arm. These points should be addressed. 

9)     The authors state: “As antibody-drug-conjugate (ADC) that is designed to combine the targeting capacity of a monoclonal antibody with the toxic properties of an established cytotoxic payload, EnaV is only using AXL as an ‘address’ to deliver its payload but does not compete with its ligand for binding [18]. As such, EnaV can exert its antineoplastic activity independently of AXL-mediated tumorigenesis. We believe this is actually an advantage for the majority of sarcomas that are displaying complex karyotypes (80-85%) and are potentially driven by multiple oncogenic pathways [10].” It is unclear what is the conclusion that maintenance of AXL activity would be an advantage based on.  Inhibition of AXL activity is not necessary for ADC activity but AXL inhibitors are being developed to treat multiple tumor types.

10)  TABLE S2- paired t-test for multiple models is labeled as “/”. It is unclear what it means, not significant?

11)  Error bars for figure 3B need to be included

12)  All data points need to be shown in figures 4A, B, C, and D.

13)  HR and “P” should be included in figure 3C.

Author Response

Comments:

  • Once the majority of tumors reached a volume of 100-150mm3, mice were randomized. Information about the range of tumor sizes should be included. Also. the criteria for selection of the three tumors per group for sacrifice at 22 days selected should be included- smaller or larger at enrolment, or size at 22 days?

The average absolute tumor volumes and ranges at start of each experiment were added to the supplementary Table S1. The following was added to the manuscript p. 3: “Three mice per group were randomly selected and sacrificed on day 22 (i.e. two weeks after last treatment) for tumor collection in 4% formaldehyde in PBS.”

  • The authors state that “The experiment ended for individual mice when the tumor volume reached >2000 mm3 or the mouse showed body weight loss >18%, with exception of three mice per group that were sacrificed earlier for histological evaluation of tumors.” However, based on the individual tumor growth curves it does not appear that these criteria were consistently used. For example, UZLX-STS126 all mice were sacrificed before reaching either one of these criteria, while UZLX-STS124 mice with control tumors were sacrificed at the tumor size of ~4000 mg.

We thank the reviewer for noticing that indeed mice of UZLX-STS126 were sacrificed earlier because of the tumor ulceration that occurs in this model when tumors reach approximately 600 – 800 mm3. The criterion ‘tumor ulceration’ was added to the manuscript (p.3), as well as to Table S3. For all other experiments, the criteria of > 2000 mm3 was met but smaller absolute tumor volume from the start in UZLX-STS124 (Table S1) can explain why several mice reached 4000% change in relative tumor volume.

  • The AXL expression on TMA was presented as an IHC score based on immunoreactivity intensity and percentage of cells at each intensity; while the AXL expression in the study tumors was stated as “based on the overall intensity of the staining: 0 - negative, 1 - weakly, 2 - intermediate and 3. Why were different criteria used?

AXL expression on TMA, performed for PDX model selection was determined by H-score, which is a semi-quantitative analysis of the target in order to compare its expression between different models. This work was performed by a certified pathologist from Genmab (P.F.). For the evaluation of ex-mouse tumours post-treatment, we evaluated the AXL expression on the whole tumor sections based on the overall intensity of the staining, a method that was successfully applied in a previous publication (https://doi.org/10.3390%2Fbiomedicines10040862). This analysis was mainly descriptive, i.e. to confirm the presence of AXL and assess possible heterogeneity of the staining.

  • Figure 1 shows the H&E of the parental tumors and the PDX models and PDX IHC. It is stated: “Original patient tumors and PDX tumors were compared and shared the same characteristic morphological features.” However, based on images in figure 1 the morphologies of the patient and PODX tissues do not seem to have similar morphology. E.g., STS3 STS126. Moreover, the H&E and AXL IHC of STS126 do not look like the same tumor. Minor comment: STS3 MDM2 IHC does not appear to be 200x

We retrieved myxofibrosarcoma patient samples and replaced pictures of H&E in Figure 2, according to comments from reviewer #1, showing now also myxoid parts of the tumors. Additionally, new pictures were taken for PDX samples of UZLX-STS126, as well as for UZLX-STS3. Furthermore, we would like to emphasize that these tumor are highly heterogeneous and variations in the extent of tumor necrosis, myxoid areas and/or cellularity might occur from passage to passage (e.g. necrosis or myxoid areas might become more present in one passage and disappear in the next). Therefore, H&E are solely used to confirm the presence of basic morphological features and is supplemented with IHC (performed on every passage of the models) and/or genetic analysis (performed on selected passages, e.g. FISH, low-coverage whole genome sequencing). For more details and data of models included in this study (with exception of the newer model UZLX-STS204), please check our publication of the STS PDX platform also referred to in the Materials and Methods: https://doi.org/10.1158/1535-7163.MCT-18-1045. Moreover, for the DDLPS models used in this project, we included MDM2 FISH performed on experimental passages in Figure 2. Additionally, the following was added/changed to the manuscript p. 4:

Despite small changes in the extent of tumor necrosis, myxoid areas and/or cellularity, original patient and PDX tumors shared the same characteristic morphological and molecular features (Figure 2).”

  • The authors state: “Additionally, a non-significant trend for tumor growth delay could be observed in the EnaV-treated tumors of the more slowly growing models STS126MFS, -STS132MFS.” Looking at the data provided, the tumors really did not grow within the first 22 days and therefore no inhibition of tumor progression was observed. It does not seem appropriate to evaluate tumor inhibition when the control tumors themselves are not growing. However, if one would be looking at day 60, the control tumor might have doubled, then this tumor could be a responder with the development of acquired resistance (day >80) (of course the concussion would be possible only if the results reach significance in powered study).

See answer 7).

  • A similar comment is valid for the survival analyses. No conclusions can be made about potential survival benefits if the differences are not significant.

See answer 7).

7) The difficulty of a correct interpretation is rather visible when one compares the 22-day tumor response of STS3, which is marked as a significant response, growth curves for the whole study of this model that does not show much inhibition, and survival analysis where the lines of control and treated groups show no differences.

Statements on non-significant trends were removed and the following was added to the discussion (p.10):

“In 5/8 STS PDX models investigated, EnaV induced a significant response in terms of tumor growth delay, tumor regression and/or improved survival compared to isotype control ADC. Although only two models responded with significant tumor shrinkage (i.e. the gold standard of response in oncology), it should be noted that tumor growth stabilization or delay is already considered a clinically meaningful response in this setting given the low objective response rates observed with doxorubicin [37]. For this reason, it would have been interesting to include a doxorubicin-treated group for direct comparison to the first-line standard-of-care, as done previously [22,23,33,38].”

8) In figure 4A tumors that did not show generally any increase in tumor volume during the 22 days period (126 and 132) have similar #/10HPF as tumors that grew relatively fast (STS3 and STS81). Moreover, the 126 showed decreased proliferation in the treatment arm. These points should be addressed. 

Because of the differences in cellularity and extend of necrosis/myxoid areas between these tumors, it is very difficult to compare the number of mitotic/apoptotic cells between models. This would only be possible when calculating a percentage of cells to compensate for more acellular or cellular areas. However, even than the number of mitotic cells might not directly correlated with the tumor growth due to role of many other factors (see for example: https://pubmed.ncbi.nlm.nih.gov/3464431/). The decreased proliferation in STS126 is noted as (p.8):

“EnaV-treated tumors of UZLX-STS3DDLPS, -STS124DDLPS, -STS204DDLPS, -STS126MFS and -STS132MFS showed a slightly reduced number of mitotic and increased number of apop-totic cells compared to isotype control ADC (Figure 4C & 4D).”

9) The authors state: “As antibody-drug-conjugate (ADC) that is designed to combine the targeting capacity of a monoclonal antibody with the toxic properties of an established cytotoxic payload, EnaV is only using AXL as an ‘address’ to deliver its payload but does not compete with its ligand for binding [18]. As such, EnaV can exert its antineoplastic activity independently of AXL-mediated tumorigenesis. We believe this is actually an advantage for the majority of sarcomas that are displaying complex karyotypes (80-85%) and are potentially driven by multiple oncogenic pathways [10].” It is unclear what is the conclusion that maintenance of AXL activity would be an advantage based on. Inhibition of AXL activity is not necessary for ADC activity but AXL inhibitors are being developed to treat multiple tumor types.

The following was changed to the manuscript (p.10): “As such, EnaV can exert its antineoplastic activity independently of AXL-mediated tumorigenesis. We believe this approach has advantages over the use of selective AXL inhibitors in is actually an advantage for the majority of sarcomas that are displaying complex karyotypes (> 80% of STS) and are potentially driven by multiple oncogenic pathways [10].”

10)  TABLE S2- paired t-test for multiple models is labeled as “/”. It is unclear what it means, not significant?

The paired t-test was performed to distinguish tumor growth delay from tumor shrinkage and thus performed only when the unpaired t-test indicated significance. An explanation for the symbol was added below the table.

11)  Error bars for figure 3B need to be included.

Error bars were added.

12)  All data points need to be shown in figures 4A, B, C, and D.

The previous figure was replaced with a figure showing individual data points.

13)  HR and “P” should be included in figure 3C.

The p-values of the log-rank test were added to Table S2.

Round 2

Reviewer 1 Report

The revision appears acceptable in the present form.

Author Response

We thank the reviewer for the time and effort that were dedicated to providing feedback on our manuscript. All comments were of great addition and significantly improved the manuscript. Thank you for your continuing interest and consideration!

Reviewer 2 Report

The authors addressed some of the reviewer's comments. However, due to the small number of animals involved in the in vivo testing and the variability of the tumor growth majority of the results were not significant, and therefore not all conclusions t seem justified. 

For example;

Statements on non-significant trends were removed and the following was added to the discussion (p.10):

“In 5/8 STS PDX models investigated, EnaV induced a significant response in terms of tumor growth delay, tumor regression and/or improved survival compared to isotype control ADC. Although only two models responded with significant tumor shrinkage (i.e. the gold standard of response in oncology), it should be noted that tumor growth stabilization or delay is already considered a clinically meaningful response in this setting given the low objective response rates observed with doxorubicin [37]. For this reason, it would have been interesting to include a doxorubicin-treated group for direct comparison to the first-line standard-of-care, as done previously [22,23,33,38].” Since the differences in tumor volumes between the control and treatment groups, the revised conclusion that there is tumor stabilization or delay in growth is not supported.

The hazard ratio was not added to the survival curves and moreover, the added Ps for the long rank test in Table S2 do not seem to correspond to the figure ( e.g., UZLX-ST204 DDLPS the survival curves overlay each other which the table show p=0.0033)

Author Response

Limited though significant tumor growth delay could be observed in UZLX-STS3 and -STS124 (Table 2). We revised our conclusions to clarify this (p.10):

“In 5/8 STS PDX models investigated, EnaV induced a significant response in terms of tumor growth delay, tumor regression and/or improved survival compared to isotype control ADC. Although only two models responded with significant tumor shrinkage (i.e. the gold standard of response in oncology), complete tumor regressions could be observed in all but one PDX tumors by the end of both experiments. An additional two models responded with limited though significant tumor growth delay (Table 2). However, it should be noted that tumor growth stabilization or delay is already considered a clinically meaningful response in this setting given the low objective response rates observed with doxorubicin [37]. For this reason, it would have been interesting to include a doxorubicin-treated group in these experiments for direct comparison to the first-line standard-of-care, as done previously [22,23,33,38].”

“In conclusion, we observed in vivo antitumor activity of EnaV in a panel of 5/8 STS PDX models investigated, at a well-tolerated dose of only 4 mg/kg. While 3/5 EnaV-responding models demonstrated either limited tumor growth delay or prolonged survival, two responded with remarkable tumor shrinkage resulting in complete tumor regressions. Moreover, AXL expression was associated with the response upon EnaV in the xenografts, although not necessarily linear with the level of expression.”

For survival analysis, p-values were corrected and hazard ratio were added to Table S2. Values were rounded to three decimal points.

We thank the reviewer for this thorough review and feedback. All comments were of great addition and significantly improved the manuscript. We hope that this revised version succeeded in offering a more clear view on the results obtained in our STS PDX.